# Digital Economy and Environmental Quality: Insights from the Spatial Durbin Model

**DOI:** 10.3390/ijerph192316094

**Published:** 2022-12-01

**Authors:** Xueyang Wang, Xiumei Sun, Haotian Zhang, Mahmood Ahmad

**Affiliations:** Business School, Shandong University of Technology, Zibo 255000, China

**Keywords:** digital economy, carbon emission, space effect, Markov chain, spatial Durbin model

## Abstract

Recent developments in attaining carbon peaks and achieving carbon neutrality have had enormous effects on the world economy. Digitalization has been considered a viable way to curtail carbon emissions (CE) and promote sustainable economic development, but scant empirical studies investigate the link between digitalization and CE. In this context, this study constructs the digitalization index using the entropy value method and spatial Markov chain, and the spatial Durbin model is employed to analyze its impact mechanism and influence on urban CE in 265 prefecture-level cities and municipalities in China from 2011 to 2017. The results indicate that: (1) The overall development level of the digital economy (DE) posed a significant spatial effect on urban environmental pollution. However, the effect varies according to the different neighborhood backgrounds. (2) The DE impedes urban environmental deterioration directly and indirectly through the channels of industrial structure, inclusive finance, and urbanization. (3) The development of the DE significantly reduces pollution in cities belonging to urban agglomerations, while the development of the DE escalates emissions in nonurban agglomeration cities. Finally, based on the results, important policy implications are put forward to improve the environmental quality of cities.

## 1. Introduction

Climate change and the excessive emission of greenhouse gases have had a huge impact on the environment and have jeopardized the well-being of this planet [1,2]. Since entering the 21st century, China’s economy has witnessed vigorous growth and has become one of the most potent economic powers in the world [3]. However, rapid economic growth leads to the unsustainable consumption of resources and the destruction of the ecological environment. The method has brought huge obstacles to China’s low-carbon transformation [4,5]. According to the British Petroleum database statistics report, as of 2020, China’s carbon emissions (CE) have reached 9.894 billion tons and are responsible for the 30.7% of global emissions [6]. Based on the frequent occurrence of ecological and environmental problems, Jinping Xi promised at the UN General Assembly that China’s “30.60” emission reduction goal would take carbon neutralization and carbon peaking as the primary purpose of ecological and environmental governance. At the same time, it will also pave the way for sustainable economic development and green and low-carbon transformation, better combine the two development concepts of eco-friendly growth and innovation drive, and better combine the two development concepts of green development and innovation-driven development, break away from the cycle of “economic development—environmental pollution”, and establish a new model of the coordinated development of environment and economy [7]. Furthermore, it has also been put forward in the “14th Five-Year Plan” that the construction of an ecological civilization will focus on carbon reduction, promote the coordinated emission reduction of atmospheric pollutants and carbon dioxide, promote the green and low-carbon development of the economy and society, and fundamentally realize qualitative change of eco-environmental improvement. It is also emphasized that the application of digital technology in production and life, to realize the digital and clean transformation of traditional industries, will help enterprises move towards low-carbon development.

The advent of the digital age has provided more ways and paths for CE reduction. Digital technology is rapidly rising and constantly changing the way of human existence due to its wide application, fast penetration, and high efficiency [8,9]. The “White Paper on the Development of China’s Digital Economy (2021)” shows that China’s digital economy (DE) development is still in a period of rapid growth in 2020, with a scale of 39.2 trillion yuan and an increase of 3.3 trillion yuan over the previous year, accounting for a 38.6% proportion of GDP and a year-on-year increase of 2.4%. This shows that digitization has become another new driving force for China’s economic growth. The impact of the DE on sustainable economic development is mainly reflected in four aspects: Firstly, the DE leads industrial development and helps build a modern industrial system [10]. Secondly, it gives full play to the enabling role of the DE and comprehensively promotes rural revitalization [11]. Thirdly, the development of the DE can effectively promote the coordinated development of regions [12]. Finally, the DE also plays an important role in promoting technological innovation [13]. The development of the DE has brought about great economic effects and far-reaching impacts on the sustainability of the environment. For example, Yu and Zhu [14] explored the mechanism of the DE on CE from the perspective of spatial heterogeneity and pointed out that the impact would be different depending on the location of regions; Zhong et al. [15] explored the impact of the application of digital technology in agriculture on CE, and the results proved that the growth of the DE can accelerate the low-carbon development of agriculture and reduce agricultural CE; Zhang et al. [16] found that the DE has become a catalyst to accelerate the transformation and development of green industries by analyzing the impact of DE development on CE in various provinces of China. This also proves that the impact of the DE on CE has been confirmed by some scholars, but there are still some deficiencies. Few existing studies take cities as research objects to probe into the impact of the DE on CE, and there is no unified conclusion on the impact mechanism or regional heterogeneity, etc.

Based on the aforementioned arguments, the following questions are raised in this paper: Will the development of the DE pose a significant impact on the CE of cities? If the answer is yes, will it enhance or decorticate environmental quality? What characteristics will it present in the space? In doing so, this study used panel data sets of 265 prefecture-level cities from 2011 to 2017 to probe into the impact of DE growth on urban CE reduction. The main contribution of this study is as follows: First, this study investigates the impact of the DE on urban CE. Indeed, research on the relationship between the DE and CO_2_ emissions is available. However, previous studies mainly focused on national and provincial data to analyze the linkage between the DE and emissions, while the literature is silent on the cities-level nexus. As per the author’s knowledge, this is the first study investigating the relationship between the DE and urban CE in prefecture-level cities. Second, this article constructs the comprehensive index of the DE at the city level and improves the measurement approach. Third, this study explores the impact of the DE growth of different types of cities on urban CE by dividing prefecture-level cities nationwide into central, eastern, and western regions and cities into urban agglomerations and nonurban agglomerations for more precise policy implications. Fourth, unlike the previous studies, this paper adopts the Markov chain, spatial Markov chain, and spatial Durbin model to study the spatial distribution and correlation between the DE and urban CE reduction to make the results more robust and accurate. Based on the results, this study provides practical policy implications to fight against climate change and achieve environmental sustainability (SDG13).

The subsequent frame structure is as follows: Section 2 is a review of the research literature on DE development and CE. Section 3 is the study design, including the selection and construction of the model methodology, variable declarations, and data sources. Section 4 is the analysis and discussion of the experimental process and results, mainly including the spatial distribution characteristics of the DE and CE, the research on the role of DE development on urban CE, and the analysis of the effects between the two. Section 5 is the conclusion and recommendations. Through the analysis and discussion of the fourth chapter, the research conclusions of this paper are drawn and policy recommendations for the future CE reduction in Chinese cities are put forward.

## 2. Literature Review

Don Tapscott first defines the economic model that presents information flow digitally as the DE in his book entitled *The Digital Economy: Hope and Crisis in the Age of Network Intelligence*. The rapid rise and wide penetration of modern information and communication technologies have spawned many new business models based on digital technologies such as the Internet and big data [17,18], and the growth of the DE has significantly promoted the reduction of greenhouse gas emissions and even the green development of industry [19,20].

The increasing greenhouse gas emissions have caused many climate disasters and threatened human survival [21]. Meanwhile, scholars have also shown solicitude for the role of the DE in CE reduction. Regarding the impact mechanism of the DE on CE, some scholars believe that the DE can reduce CE by changing the regional industrial structure and increasing the proportion of the tertiary industry [22,23]. For instance, from the perspective of the development of the DE, Dong et al. [24] considered the win–win issue of economic development and CE reduction. They pointed out that industrial structure adjustment has an obvious inhibitory effect on CE, and there is also a one-way Granger causality. Some scholars believe that the DE can affect CE by affecting the energy consumption structure [25], including Li et al. [26]. Through the verification of the influence of the energy-resource structure and DE on CE, it was pointed out that the energy-resource structure dominated by coal has a great influence on CE. In addition, some studies have also indicated the impact of resource factor allocation on CE in the process of DE development [27,28], and Xu et al. [29] calculated China’s digital technology investment from 2001 to 2018 and explored the impact of digital capital flows on CE. The results showed that digital capital has an important role in promoting CE efficiency. Moreover, it also has a spatial spillover effect, which can promote CE efficiency in surrounding areas. Wang et al. [30] used a panel data set of Chinese cities, trying to provide a more systematic perspective on the relationship between the DE and urban low-carbon development, and also pointed out that the flow of innovative elements was the key communication channel for the DE to affect the low-carbon sustainable development of cities. Han et al. [31] used the SBM-ML index to measure regional total factor carbon productivity (TFCP), and a threshold model was constructed to explore the impact of the DE on TFCP under different technology accumulation thresholds. The results showed that with the improvement of the technology accumulation level, the effective coefficient of the DE on TFCP continued to increase, and the significance level was increasing. Most studies have pointed out the advantages of the DE for CE reduction, but its negative effects should also be paid attention to. The DE not only promotes economic development but also expands the demand for energy consumption, thus increasing CE. This also shows that the impact of the DE on CE is not only a simple linear relationship. Han et al. [32] and Li [33] also confirmed this through research. There is a nonlinear relationship between DE and CE, and it is pointed out that the impact of DE on CE presents an inverted U-shape.

The growth of the DE will be accompanied by the progress and innovation of digital technology, which has a considerable impact on CE [34,35]. Kwon et al. [36] and Shin [37] believed that the use and innovation of information technology contribute positively to environmental governance. The application of information technology can improve the accuracy and efficiency of relevant data and scenario analysis, helping environmental pollution control. The impact of digital technology on CE will have different results for different regions and different scenarios [38]. Su et al. [39] verified the effect of digital technology innovation on the CE intensity of BRICS countries by quantifying information and communication technology. The results showed that most innovative tools increase CE, and the use of mobile networks reduces CE. Likewise, Wang et al. [40] reached a similar conclusion. Based on the KPWW method and multi-panel regression, the impact mechanism of digital technology innovation on CE was explored. The results also showed duality. On the one hand, digital technology innovation will increase CE, and on the other hand, cross-industry technology spillovers will continue to reduce CE. In addition to the duality shown above, the impact of digital technology innovation on CE will also show a unilateral effect [41,42]. Through the analysis and exploration of the future energy CE path of digital technology, Chen et al. [43] proposed accelerating the innovation and development of carbon reduction technology and applying digital technology globally to accelerate the efficiency of CE reduction. Ma et al. [44] incorporated the DE and trade-adjusted CE into the same research system. Based on the development level of the DE, he evaluated China’s trade-adjusted CE and found that the DE has a negative impact on consumption-based emissions.

With the continuous development of human socioeconomic activities and urbanization, cities have become the most carbon-intensive areas. With its technological advantages, the DE provides technical support for the city to reduce CE by designing a scientific and reasonable trading mechanism and building a relevant trading platform [45]. In the process of economic development, the development of the DE will also adjust the level of innovation and industrial structure, etc., to make it develop in a low-carbon direction [33]. Carbon dioxide has a certain mobility, so it will have an impact on the surrounding areas; thus, space research is necessary. Zhang et al. [42] used the SDM model to verify that China’s DE development can effectively curb urban CE and pointed out that urban energy consumption, energy intensity, and green area are the main impact ways. Wang et al. [46] also pointed out the emission reduction effect of the DE and found that this spatial impact would be affected by the number of industries through heterogeneity analysis. The development of the DE will inevitably be accompanied by the construction of information infrastructure. In this process, it will promote the efficient allocation of resource elements, enhance technological innovation and achievements transformation, thereby reducing urban CE [47], and will drive the development of digital finance. Digital finance has natural green attributes and technological advantages, which can effectively solve the financing problems of green and low-carbon industries and promote the development of low-carbon industries, making urban CE effectively controlled [48].

At present, most of the relevant studies on the CE of the DE are mediated by a certain element, while there are few relevant studies on the direct impact of the DE on CE. Moreover, although the development of the DE will affect CE from many aspects, the specific impact effects need to be further analyzed. Therefore, taking urban CE as the research object, this paper explores the spatial effect and spatial characteristics of the DE on urban CE to provide experience and reference for urban CE reduction with the background of the DE.

## 3. Research Design

This paper aims to explore the spatial impact of DE development on urban CE. Therefore, this study first evaluated the development of the DE with the help of the entropy method, then used the Markov chain to predict the hierarchical transfer of CE, and finally used the spatial Durbin model to conduct a series of empirical studies. The following is an introduction to the methods used, as well as a statement of the data sources and variable selection.

### 3.1. Research Methods

#### 3.1.1. Entropy Method

The entropy value method mainly uses the characteristic that entropy is an uncertain measure to judge the validity and value of the existing indicators. The size of the entropy value can be expressed as the extent of disorder of a system. If the extent of the order of the system is higher, the entropy value of the system will be larger, and the amount of information contained will be less; in contrast, the higher the disorder level of the system, the smaller the entropy of the system and the more information it contains. Therefore, the entropy method can be used to measure the amount of information in each index in the evaluation index system and determine each index’s weight. The specific construction steps are as follows:

Normalize the data, which represent the row and the column data in the overall data:(1)$$x_{\mathrm{ij}}=\frac{x_{\mathrm{ij}}-\min\left(x_{j}\right)}{\max\left(x_{j}\right)-\min\left(x_{j}\right)}$$

Calculate the proportion of the record under index:(2)$$P_{\mathrm{ij}}=\frac{x_{\mathrm{ij}}}{\sum_{1}^{n}x_{\mathrm{ij}}}$$

Calculate the entropy of index:(3)$$e_{j}=-k\times \sum_{1}^{n}P_{\mathrm{ij}}\times \log\left(P_{\mathrm{ij}}\right)$$

Among them $k=\frac{1}{\ln\left(n\right)}$, and assuming that when $P_{\mathrm{ij}}=0\;$equal to 0, $P_{\mathrm{ij}}\times \log\left(P_{\mathrm{ij}}\right)=0$.

Calculate the difference coefficient of index j:(4)$$g_{i}=1-e_{j}$$

Calculate the weight of index j:(5)$$W_{j}=\frac{g_{j}}{\sum_{1}^{m}g_{j}}$$

Calculate the combined score of the ecosystem:(6)$$F_{i}=\sum_{i=1}^{n}W_{j}x_{\mathrm{ij}}$$

#### 3.1.2. Markov Chains

The Markov process refers to classifying the continuous attribute values of geographical phenomena at different times by grades and converting the data into different k types by discretization, to calculate the probability distribution of various types and their changes to reflect the evolution law of things. This paper applied this method to reflect the evolution law of urban CE levels. Specifically, the continuous urban CE level data was first discretized into k types, and then the probability distribution of different types of transitions was calculated to reflect the evolution process of urban CE levels. Usually, the probability distribution of urban CE level type in year t is expressed as a 1×k state probability vector${\;P}_{t}$, denoted as $P_{t}$ = [$P_{1t}$, $P_{2t}$,… $P_{\mathrm{kt}}$], and the transfer between urban CE levels in different years can use a k×k, which is represented by the Markov transition probability matrix M, where the element $m_{\mathrm{ij}}$ represents the probability that a city belonging to type i in year t will transfer to type j in the next year. In this paper, all cities were divided into four types by the natural segment point method—k =1, 2, 3, 4, which are used to represent, respectively, k, the larger the city, the greater the CE intensity of the city. A transition of state type from high intensity to low intensity is defined as a downward transition, and vice versa as an upward transition. Adopting the following formula estimates:(7)$$m_{\mathrm{ij}}=\frac{n_{\mathrm{ij}}}{n_{i}}$$
where $n_{\mathrm{ij}}$ represents the total quantity of areas pertaining to type i at time t and transferred to type j at time t+1 over the whole period; $n_{i}$ is the total quantity of areas pertaining to type i in all years of transfer during the period.

#### 3.1.3. Spatial Markov Chains

Although the traditional Markov chain method can be used to analyze the evolution of regional convergence (differentiation), it is difficult to reveal the spatial characteristics of regional convergence because different regions are regarded as “islands”, and the spatial interaction between regions is ignored. The spatial Markov transition matrix considers the influence of the neighborhood background. By establishing the Markov chain transition probability matrix under different spatial lag conditions and analyzing the convergent evolution law of the regional unit under different neighborhood backgrounds, it can simultaneously analyze the spatial and temporal aspects. Based on the city scale, the traditional Markov chain is decomposed into a k k×k conditional transition probability matrix based on the spatial lag type (type) of the k city i in the year t. For the first k conditional matrix, the elements $m_{\mathrm{ij}}\left(k\right)\;$represent the spatial transition probability that the city belongs to type i and is transferred to type j in the next year under the condition of the spatial lag type k of the city in the year t. The specific formula is as follows:(8)$${\mathrm{Lag}}_{a}=\sum Y_{b}W_{\mathrm{ab}}$$
where $Y_{b}$ represents the attribute value of a city unit and $W_{\mathrm{ab}}$ represents the spatial weight matrix. This paper determines the spatial weight through the common boundary: ${W\;}_{\mathrm{ab}}=1$ indicates that the city a and the city b have a public boundary, $W_{\mathrm{ab}}=0$ means that city a and city b have no public boundaries.

#### 3.1.4. Spatial Measurement Model

Due to the economic and production links between cities, the CE of each city does not exist in isolation but is bound to be spatially correlated with other cities. When exploring the influencing factors of CE, to improve the estimation results for accuracy, this spatial correlation must be taken into account. Therefore, a spatial econometric model needs to be established. In this study, the spatial Durbin model was constructed as follows:(9)$${\mathrm{lnCO}}_{2_{\mathrm{it}}}=\rho \sum_{j=1}^{n}W_{\mathrm{ij}}{\mathrm{lnCO}}_{2_{\mathrm{it}}}+\alpha_{0}+\beta_{1}X_{\mathrm{it}}+\sum_{j=1}^{n}W_{\mathrm{ij}}X_{\mathrm{it}}\gamma_{1}+\mu_{i}+\lambda_{t}+\varepsilon_{\mathrm{it}}$$

Among them, ${\mathrm{lnCO}}_{2_{\mathrm{it}}}$ represents the urban CE level; X represents the explanatory variable, including the core explanatory variable DE measurement level and other control variables; ρ represents the spatial lag regression coefficient, reflecting the degree of mutual influence of CE in spatially adjacent areas; $\alpha_{0}$ represents the constant term; $\beta_{1}$ represents the regression coefficient of the $\gamma_{1}$ explanatory variable; $\gamma_{1}\;$represents the spatial lag regression coefficient of the explanatory variable; $\mu_{i}$ represents the regional effect; $\lambda_{t}$ represents the time effect; $\varepsilon_{\mathrm{it}}$ is a random interference term. Figure 1 shows the method connection and empirical application.

### 3.2. Variable Description

#### 3.2.1. Explained Variable

This paper explores the effect of the development level of the DE on urban CE. At present, there are two main choices for explanatory variables in the research on CE reduction: one is per capita CE [49,50] and the other is total emissions [51,52]. This paper selected the total urban emissions as the explained variable in this study. The per capita CE were not selected because there is a large population flow in cities every year. If the per capita emissions were selected, the uncertainty of the variables would increase in the experiment error in the result.

#### 3.2.2. Explanatory Variables

This paper intends to measure the development level of the DE from five aspects: Internet penetration rate, related business output, industry employees, mobile phone penetration rate, and digital finance development. Following the previous studies [53,54], this paper selected the relevant indicators in the process of DE development and constructed the DE development indicators using the entropy method. See Table 1 for the specific index construction.

### 3.3. Control Variable

At present, there have been plenty of studies on the influencing factors of CE intensity in the literature [55,56,57]. Based on previous related research, this study selected the following variables as control variables:

Economic growth (lnGDP): Based on the environmental Kuznets theory, the level of CE will continue to increase with the degree of economic development and decrease with the improvement of economic development after reaching a certain level. Many scholars have also confirmed this theory, so there is no doubt that the level of economic development impacts CE [58,59]. This paper takes the natural logarithm of GDP per capita by following the previous study of Wang et al. [60].

Industrial structure (IS): The secondary industry includes mining, manufacturing, electricity, gas, and other highly polluting and high-emission industries, while service industry dominates the tertiary industry. The proportion of the secondary industry affects the degree of environmental pollution and CE. Therefore, whether the secondary or tertiary industry dominates the industrial structure is related to carbon dioxide emission intensity. This is calculated by the ratio of the tertiary industry to the secondary industry [61].

Foreign investment (FDI): Foreign investment is a double-edged sword. For a region, foreign investment can bring a lot of capital, technology, and new development models, which are all conducive to promoting technological progress and reducing CE. However, while introducing capital and technology, foreign companies will transfer some high-polluting, high-emission, low-tech companies to the region, leading to an increase in CE. FDI is derived from the ratio of foreign investment to GDP [62].

Urbanization (UB): The advancement of urbanization can effectively improve the economic level and technological innovation, and both economic development and technological progress will have a certain impact on CE. In addition, urbanization will also change the industrial structure and energy consumption structure. This is also an important way to curb CE effectively. It is represented by the proportion of the urban population to the total population [63].

Energy consumption (EC): Energy consumption from fossil fuels (i.e., coal, oil, and natural gas) is considered the main source of carbon dioxide, and changes in energy consumption will inevitably lead to changes in CE. In this paper, various energy sources are converted into standard coal to calculate CE [64]. The specific variable definition is shown in Table 2.

### 3.4. Study Area and Data Source

As the most active and intensive region of human socioeconomic activities, cities have become the regions with the most concentrated CE, and urban emission reduction is of major implications for achieving global emission reduction goals [65,66]. Therefore, cities are regarded as the implementation object of emission reduction measures in most areas, and they strive to promote the goal of building a low-carbon city. This paper takes cities as the basic unit of research, including 265 prefecture-level cities and municipalities across the country, and the time span is 2011–2017.

On the basis of the data’s professionalism, rigor, and objectivity, this study built panel data sets of 265 prefecture-level cities in China from 2011 to 2017. The data used in this study to construct the DE development level indicators and the control variable data come from the *China Urban Statistical Yearbook*, the *Prefectural-level City Statistical Yearbook*, and the National Bureau of Statistics. The carbon emissions data used in this study are from China Carbon Accounting Databases (CEADs). The specific website is: https://www.ceads.net.cn. The descriptive statistics of the data are shown in Table 3.

## 4. Results and Discussion

### 4.1. Spatial Distribution Characteristics and Correlation Analysis of the DE and CE

Before exploring the spatial spillover effects and influencing factors, this paper analyzes the spatial distribution characteristics and spatial correlation of the urban DE development level and CE intensity. As shown in Figure 2, this paper selects the development level of the urban DE in 2011, 2014, and 2017 to discuss. With this, the spatial distribution of the development level of the DE has gradually evolved from a “multi-point” scattered distribution to a “cluster” aggregation distribution, and the DE level of many cities has been significantly improved. During the period from 2011 to 2014, cities noticed the development prospects of the DE and started to catch up with development. However, owing to the limitations of resource endowments, location factors, and other factors in various regions, the regional differences have not narrowed but have a tendency to expand, and they have formed a distribution pattern with the Beijing–Tianjin–Hebei and Chengdu—Chongqing urban agglomerations, the Yangtze River Delta, and other urban agglomerations as the core. Since 2014, the development level of DE development in various regions has been significantly improved, and a group form has been formed in which the core cities of the DE spread to surrounding cities. However, due to the siphonic effect of the core city, the difference between it and the surrounding cities also increases.

Regarding the spatial distribution characteristics of urban CE intensity, from Figure 3, the spatial pattern of CE shows the characteristics of low in the south and high in the north, and there is a large difference between the north and the south. This is mainly because the main source of heating heat and electricity in the north comes from various types of coal. In the southern region, the Chengdu–Chongqing urban agglomeration and the Yangtze River Delta have become the main emission areas. These two regions are economically developed and have a high level of industrialization, and industrial emissions are the main source of CE, which is also the main reason why Chengdu–Chongqing and the Yangtze River Delta have higher emissions than other southern cities. On the whole, the maximum value of CE decreased from 220.745 in 2011 to 192.504 in 2017, and the value gap showed a decreasing trend, which shows that China’s low-carbon policy has certain effects on cities with high emission levels.

To further explore the spatial characteristics of urban CE in China, this study adopted the global Moran’s I index to describe the spatial signature of China’s urban CE and check the spatial autocorrelation. Table 4 shows the changing trend of the global Moran’s I index during the whole study period. All Z values exceed 7 and pass the significance test, which indicates that China’s CE intensity has obvious spatial clustering characteristics. As time progresses, Moran’s I index shows a trend of first falling and then rising, but on the whole, the degree of spatial agglomeration is weakening—this is because the degree of the close connection of various regions in economic development will have an impact on the degree of the spatial agglomeration of CE. In addition, the development of society, the advancement of urbanization, and changes in the natural environment will lead to the spatial autocorrelation of urban CE.

### 4.2. Dynamic Transfer Analysis of Urban CE

#### 4.2.1. Dynamic Transfer of Urban CE under Unconstrained Conditions

To analyze the transfer dynamics and state transfer vitality of urban DE development, this study divided the CE intensity of each city into four dimensions: low, medium, sub-high, and high according to the natural breakpoint method, corresponding to K = I, II, III, IV. Furthermore, the method of the Markov chain was used to calculate the transition probability of CE in each region from 2011 to 2017. The transition probability is shown in Table 5. On the whole, the CE intensity of each level has an obvious convergence effect. The main diagonal value reflects the probability that the urban CE level remains unchanged. The maximum value is 97.7% and the minimum value is 97.0%, that is to say, the minimum probability that the urban CE level remains unchanged during the study period is 97.0%, which also shows that the CE intensity of the city is not easy to transfer and has strong effect stability. Simultaneously, it should give heed to the “Matthew effect” of urban CE. Taking the main diagonal as the boundary, the lower part is the probability of CE intensity shifting downwards. The probability of maintaining the original level unchanged in areas with a low initial level is 97.7%, and the probability of further reduction is 2.3%, which indicates that such cities may have entered the late stage of industrial transformation, and the emission reduction contribution made through industrial transformation has been saturated. To further reduce the emission intensity, it needs to start from other aspects. The probability of keeping the original level unchanged is 97.2% in the region with the initial level of high intensity, and the probability of shifting to sub-high intensity is only 2.8%, indicating that the region may be trapped in a resource-dependent and path-locked development path, which makes it difficult to improve energy efficiency. From the perspective of the probability of transition from each level to the next level, it is very difficult to reduce the level, and there is only a single level spanning and no multi-level spanning. This also shows that the process of CE reduction is a step-by-step process. To achieve multi-level transition CE reduction, it is essential to further strengthen CE reduction efforts. In addition, through Figure 4, the transition of emission levels in Chinese cities can be more intuitively seen. During the entire study period, only 3.77% of the cities experienced a level decline, and the probability of a level rise was even lower at 1.13%. Still maintaining the same level of emissions, it can be inferred from the analysis that the CE intensity of Chinese cities has been effectively controlled, but there has not been a significant drop in urban emissions.

#### 4.2.2. Dynamic Transfer Analysis of Urban CE under the Constraints of Geographic Neighborhood

The CE intensity of Chinese cities is not independent of each other in geographic space. The CE intensity of cities is often affected by the region where they are located and has strong spatial agglomeration and spatial interaction effects. To analyze the impact of the geographical environment on the transfer of urban CE types, this study added the neighborhood type as a condition on the traditional Markov transition probability matrix and obtained the spatial Markov transition probability matrix to explore the effect of CE intensity levels. The results of the transition probability under the influence of adjacent regions are listed in Table 6. After different regional backgrounds are used as conditions, the state transitions of regions show great differences, so the effect of the regional background on the state transition of regions is significant. When an area has low intensity as its neighborhood, the probabilities of the main diagonal are 97.7%, 93.3%, 100%, and 100%, respectively. It is not difficult to see that the area with medium emission intensity is affected the most and the probability of falling to a low level is 5.9%, and the neighborhood state significantly impacts the emission intensity. From the perspective of the emission intensity transition probability of the adjacent areas of the middle-level emission intensity, the high-intensity area will be significantly affected, and the probability of falling to the sub-high level is 16.7%, and there is a certain probability that the region with sub-high intensity and medium intensity will decline in level. Among the cities with the sub-high level as the neighborhood, the cities with the low level as the initial level have sub-high CE activity, with an 8.6% probability of rising to the medium-intensity level, while the medium, sub-high and high levels have strong stability. There is no state transition in cities with high intensity, but the number of samples in such cities is small, and the conclusion is a lack of reliability.

### 4.3. Influence of DE Development on Urban CE Intensity

#### 4.3.1. Spatial Measurement Model Selection

It is imperative to use an appropriate model to accurately assess the CE intensity of the development of the DE. In this paper, the LM and R-LM tests were firstly carried out. Both test results were significant at the 1% level; that is to say, the spatial error term and the spatial lag term exist simultaneously, so this study chose the spatial econometric model for regression analysis. This paper carried out the LR and Wald tests for the spatial econometric model, including SDM, SEM, and SLM; the results showed that they all passed the 1% significance test, indicating that the SDM model is not likely to degenerate into SLM and SEM. Thus, the original hypothesis of SEM and SLM model was rejected. SDM is used as a spatial measurement model to explore the impact of DE development on urban CE intensity. Finally, the Hausman test was carried out, and the test results were also significant at the 1% level, so this study chose a fixed-effect model for estimation. The specific test results are given in Table 7.

#### 4.3.2. Spatial Measurement Model Selection

To find out the impact of DE growth on urban CE under different spatial conditions, this paper constructed three spatial weight matrices: adjacency weight matrix (W1), geographic weight matrix (W2), and economic distance matrix (W3). Table 8 shows the regression results of the spatial Durbin model under the three weight matrices. From the table, the parameters of the DE in the SDM regression under the three weights are −0.078, −0.129, and −0.243, and the results all passed the 1% significance test, which means that regardless of the spatial conditions, it turns out the development of the DE has effectively reduced CE. The development of the DE has driven the industrial structure optimization and also promoted the energy structure. The general application of digital technology has improved the efficiency of resource allocation, which has further optimized industrial emissions and urban energy efficiency, thereby reducing urban CE. In terms of the impact of control variables on CE, the level of economic development (lnGDP) has an obvious promoting effect on urban CE in the SDM regression under the adjacency weight and economic weight, but the economic development has no impact on CE under the geographical weight. In terms of industrial structure (IS), the development of the DE under the three weights hurts CE and has passed the significance test at the 1% level. Most industries in the secondary industry have made huge “Contributions” to CE. The development of the DE has promoted the gradual transfer of the industrial structure from the secondary industry to the tertiary industry, while also reducing the intensity of CE. Foreign investment (FDI) and energy consumption (EC) all show significant promoting effects on CE under the three weights. The investment will bring a lot of capital and new technologies, but at the same time, it will also bring low-tech, high-polluting, high-emission enterprises into the city, which will inevitably increase regional CE. At present, China’s main power generation methods and heating sources are still coal-fired, gas-fired, and oil-fired boilers, all of which require a large amount of coal and will inevitably increase the urban CE. Under the three spatial weights, urbanization (UB) hurts CE and is significant at the 1% level, which also means that CE will be significantly reduced with the advancement of urbanization. Urbanization will promote economic growth. According to the Petty–Clark law, when the social income level increases, the employed population will transfer from the primary industry to secondary industry; when the social income level further increases, the employed population will flow from the secondary industry to tertiary industry. The increase in the tertiary industry can effectively reduce CE. Urbanization also has a technological effect. Urbanization will promote technological innovation, improve production efficiency and energy efficiency, and, in turn, reduce CE.

#### 4.3.3. Influence Effect Analysis

To have a clearer understanding of the impact of the interior and exterior of the city on CE, this paper analyzed the total effect, direct effect, and indirect effect of the estimated model. Table 9 shows the specific results of the effect analysis. For the development of the DE, whether it is an indirect effect or a direct effect, the growth of the DE has an obvious inhibition effect on urban CE. Within the city, the growth of the DE has promoted the industrial structure. With the improvement of the energy consumption structure, the application of information technology has greatly enhanced the efficiency of resource allocation. These are important factors to reduce CE effectively. With the continuous advancement of informatization and intelligence, the connection between cities is gradually increasing the interaction between technology and resources, which also means the CE of neighboring cities is affected by the development of the DE in the city, and the reduction of CE in neighboring cities will, in turn, affect the emissions of the city. This will further suppress the urban CE and form a benign cycle. From the perspective of indirect effects, the development of the DE has a spatial spillover effect. Every 1% increase in the level of the DE will have a 4.509% impact on the CE of surrounding cities. The direct effect coefficient and indirect effect coefficient of economic development were both significantly positive, 0.083 and 2.112, respectively. That is to say, the economic development of the city will lead to the aggravation of CE, and the economic development of neighboring cities will also exert a positive influence on the urban CE. The direct and indirect effects of the industrial structure are significantly negative. The transition of the industrial structure from the secondary industry to the tertiary industry has reduced energy consumption, which will naturally reduce urban CE. Foreign direct investment shows different effects in direct and indirect effects. From the perspective of direct effect, foreign investment significantly increases the intensity of urban CE, but foreign investment in adjacent areas will reduce CE in the region, and the reduction effect was significantly higher than the promotion effect of the city; thus, the total effect showed a significant inhibitory effect. From the direct effect of urbanization, its regression coefficient is −0.517 and passed the 1% significance level test. Urbanization has the effect of promoting economic development and scientific and technological progress. With the continuous progress of urbanization, the economy and technology are also improved. According to the environmental Kuznets theory, after the economy reaches a certain level, the increase in per capita income will reduce the environmental impact and gradually improve environmental quality. The direct and indirect effect coefficients of energy consumption are both significantly positive. The burning of fossil energy is the main source of CE. The emission of carbon grows in number with the enhancement of energy consumption. Due to the spatial spillover effect, energy consumption in adjacent areas will also produce an effect on urban CE.

### 4.4. Spatial Heterogeneity Analysis

#### 4.4.1. Regional Heterogeneity Analysis

Considering the spatial heterogeneity of the impact of DE development on urban CE, this study adopted the approach of Liu and Dong [67] to divide Chinese cities into eastern, central, and western parts to explore the effect of DE growth on CE in cities located in different locations. Because the influence of spatial heterogeneity is mainly considered, follow-up research is based on the geographic weight matrix to expand the estimation. Table 10 shows the specific results of the heterogeneity analysis. The DE growth of eastern cities has no impact on urban CE. This may be because the eastern region began to develop the DE earlier with its good resource endowments and location advantages, which has effectively restrained the CE of cities. However, with the passage of time, the inhibitory effect showed a downward trend until the stable level. It is manifest in Figure 2 and Figure 3 that CE in the eastern region does not change significantly throughout the study period and has been maintained at a certain level, which is also the main reason why the DE growth of eastern cities shows no significant effect on urban CE. In the central region, the regression coefficient of DE development is 0.351, which is very significant. That is to say, the development of the DE has significantly promoted CE. The core cities in the central region have obvious advantages. During the development process, the core cities gather various elements due to their siphon effect, which makes the development of the cities outside the core lag behind, and it is difficult to exert the spillover effect of the development of the DE. This also leads to CE in the cities outside the core that cannot be effectively alleviated. Secondly, most of the central cities exchange energy consumption for economic development, which is also one of the reasons for the high CE. In the western region, the regression coefficient of the DE is −0.408 and significant at the 1% level. Compared with the central and eastern cities, the development is relatively backward, but this has also become a major advantage in the western region. Relying on the relatively mature DE development models and experience in the central and eastern regions, the development efficiency of the DE can be fully improved and supplemented by appropriate development policies and the introduction of a large number of funds, personnel, and technologies, so that the DE in the western region can attain a high level. It has a higher improvement and also has the effect of curbing CE.

#### 4.4.2. Heterogeneity Analysis of Urban Agglomerations

This paper also discusses the relationship between the DE and CE from the perspective of urban agglomerations and selects five representative urban agglomerations in China, namely the Pearl River Delta urban agglomeration, the Yangtze River Delta urban agglomeration, the urban agglomeration in the middle reaches of the Yangtze River, the Chengdu–Chongqing urban agglomeration, and Beijing–Tianjin–Hebei urban agglomeration. From the results in Table 11, the regression coefficient of the DE of the cities belonging to the urban agglomeration is −0.125, and it has passed the 1% significance level test, which shows that the DE development of the cities in the urban agglomeration will significantly reduce the CE. Within the urban agglomeration, the siphoning effect is very obvious, and it draws a lot of resources from surrounding cities to develop itself. The development level of the DE differs greatly from that of external cities, and the dividends of the DE are released more fully, so the inhibition of CE is more significant. Nonurban agglomeration cities are on the contrary. The regression coefficient of the DE in nonurban agglomeration cities is significantly positive, which means that the development of the DE in nonurban agglomeration cities will accelerate the CE of cities. The urban agglomeration cities have lost a lot of resources, the economic development is slow, and the industrial structure is still dominated by secondary industry. The lack of resources makes the efficiency and quality of the development of the DE lose its guarantee. The slow development of the DE also makes the optimization and upgrading of the traditional high energy consumption industry and the optimization of the energy consumption structure lose a great help, which all accelerates the urban CE and further aggravates the urban CE.

### 4.5. Impact Mechanism Analysis

The results of spatial measurement fully reflect the inhibition effect of the DE on urban CE. To further explore the path of the DE to inhibit urban CE, this section analyzes the mechanism from the perspective of industrial structure, inclusive finance, and urbanization. The following constructs the intermediary effect test model:(10)$${\mathrm{IS}}_{\mathrm{it}}\left(\mathrm{IF}、\mathrm{UB}\right)=\theta_{0}+\theta_{1}{\mathrm{DEI}}_{\mathrm{it}}+\theta_{2}X_{\mathrm{it}}+\lambda_{t}+\mu_{i}+\varepsilon_{\mathrm{it}}$$
(11)$$\mathrm{lnCO}2_{\mathrm{it}}=\gamma_{0}+\gamma_{1}{\mathrm{DEI}}_{\mathrm{it}}+\gamma_{2}{\mathrm{IS}}_{\mathrm{it}}\left(\mathrm{IF},\mathrm{UB}\right)+\gamma_{3}X_{\mathrm{it}}+\lambda_{t}+\mu_{i}+\varepsilon_{\mathrm{it}}$$

According to the stepwise regression method of the intermediary effect test, when the regression coefficient of Equation (10) is significant, the subsequent test shall be conducted. When the coefficient of Equation (11) is significant at the same time, there is a mediation effect. If the regression coefficient of Equation (11) is also significant, this is a partial intermediary effect; otherwise, it is the full intermediary effect. However, in recent years, the step-by-step method has been heavily questioned [68,69]; thus, this paper added the bootstrap method to further verify the significance of product coefficients, to make the test results more accurate. See Table 12 for specific inspection results.

The results of the mechanism tests are reported in Table 12. From columns (1), (3), and (5), it observes that the regression coefficient of the DE for industrial structure, inclusive finance, and urbanization has passed the significance test. That is to say, the growth of the DE has improved the industrial structure, promoted the development of inclusive finance, and promoted the process of urbanization. It can be inferred from columns (2), (4), and (6) that after considering the intermediary variables, the regression coefficient of the DE for CE is still significantly negative, and the regression coefficients of the industrial structure, inclusive finance, and urbanization are −0.1298, −0.0008, and −0.3213, respectively, and the bootstrap result is also significantly negative, which proves that the growth of the DE has an intermediary effect on urban CE through the industrial structure, inclusive finance, and urbanization. In addition, it can be noted that the regression coefficient between the industrial structure and urbanization is significantly greater than inclusive finance, which means that the industrial structure and urbanization are the main and most effective ways to reduce urban CE in the DE.

## 5. Conclusions and Suggestions

### 5.1. Conclusions

This study analyzed the spatial evolution characteristics of the DE and CE and further probes into the effect of the DE on the spatial effects of urban CE impacts. Using panel data sets from 265 cities in China, the development index of the DE was constructed by the entropy method, and the relationship between the DE and CE was explored through spatial econometric models and heterogeneity analysis. The specific research results are as follows:Through the analysis of the development level of the DE and the spatial distribution characteristics of the urban CE intensity, it is manifest that the growth level of the DE has been strongly improved during the entire research period. Compared with the overall development level, it has raised a gradient, gradually evolving from a “multi-point” scattered distribution to a “clustered” distribution. From a local point of view, the differences between regions are obvious, and there is an expanding trend, forming a pattern with Beijing–Tianjin–Hebei, the distribution pattern of the Chengdu–Chongqing urban agglomeration, the Yangtze River Delta, and other urban agglomerations as the core. The overall CE of the city has been significantly reduced. The maximum emission value has been reduced from 220.745 to 192.504. The spatial distribution shows obvious north–south differences. The emissions in the south are far lower than those in the north, but the emissions in Chengdu and Chongqing and the Yangtze River Delta are particularly prominent.The transition probability of urban CE intensity was studied by using the Markov chain and spatial Markov chain methods. In the research on the dynamic transfer of urban CE under unconstrained conditions, it can be found that cities at each level have strong stability and are not prone to changes in levels, and the cities with declining emission levels only accounted for 3.77% of the total. In cities with declining levels, there is only a single level decline, and there is no cross-level decline, which also shows that the process of CE reduction is gradual. To achieve the emission reduction of multi-level transition, it needs to strengthen the intensity of CE reduction further. The research on the dynamic transfer of urban CE under the constraints of geographic neighborhoods found that the state transfer of regions shows great differences under the conditions of different regional backgrounds. When a region takes low intensity as its neighborhood, the region with medium emission intensity is the most affected, and the probability of falling to a low level is 5.9%. From the perspective of the emission intensity transition probability of the adjacent areas of the middle-level emission intensity, the high-intensity areas will be significantly affected, and the probability of falling to the sub-high level is 16.7%, and the areas with sub-high and medium intensity have a certain probability of descending the hierarchy. Among the cities with the sub-high level as the neighborhood, the cities with the lower level as the initial level have sub-high CE activity, with a probability of 8.6% rising to the medium-intensity level, while the medium, sub-high, and high levels have strong stability.This study analyzed the spatial effect of DE development on CE through spatial measurement. The results show that under the three spatial weights, the regression coefficients of the DE on urban CE are −0.078, −0.129, and −0.243, respectively, and the results have passed the 1% significance test. This shows that the development of the DE will have an inhibitory effect on urban CE regardless of spatial conditions. In the effect analysis, the regression coefficients of the direct effect, indirect effect, and total effect are all significantly negative, and the coefficient of the indirect effect is much larger than the direct effect, which means that the development of the DE in this city will reduce the urban CE. The adjacent development of an urban DE will also reduce the urban CE, and the reduction effect is much higher than the reduction effect of the local urban DE development on the urban CE.The results of the heterogeneity test of the impact of the DE on CE are: the development of the DE in the eastern region has no impact on CE in cities, the development of DE in central regions promotes urban CE, and the development of the DE in western regions will effectively reduce the urban CE. From the perspective of whether it is an urban agglomeration, the regression coefficient of the DE of the cities belonging to the urban agglomeration is −0.125, and it is significant, which proves the CE of the cities in the urban agglomeration will gradually decrease due to the development of the DE. On the contrary, the DE regression coefficient of the cities in the periphery of the core urban agglomeration is 0.128. The development of the DE will promote urban CE. The different results of the two types of cities also reflect the spatial complexity of urban CE. The results of the impact mechanism point out that the DE can reduce urban CE by improving the industrial structure, promoting the development of inclusive finance, and accelerating the process of urbanization, and the industrial structure and urbanization are the main ways.

### 5.2. Policy Suggestions

The main conclusion of this paper is the research on the current specific national conditions in China, and on this basis, it expands the research angle of carbon emission influencing factors and supplements the theory of empirical research evidence. Therefore, it has more prominent practical significance. In addition, for other developing countries or regions with relatively backward economic development in the world, their digital economy development status is similar to that of China, so the research conclusions and policy recommendations of this paper can also provide useful lessons for them. Specific suggestions are as follows:Accelerate digital technology innovation and strengthen digital system construction. First, promote digital technology innovation, improve the innovation efficiency of artificial intelligence, cloud computing, blockchain, big data, and other digital technologies, expand the application breadth and depth of digital technology, and accelerate the practical application of innovation achievements. Then, formulate supporting policies for the development of low-carbon industries driven by the DE. Combined with the characteristics of low-carbon industry development, improve relevant supporting policies, and improve the low-carbon industry system and digital standard specification system. Strengthen financial investment in the development of digital low-carbon industries, increase incentives for the digital development of low-carbon industrial chains, optimize the development environment for low-carbon industries, and actively bring into play the positive and dynamic role of digital technology.Increase capital investment to promote digital infrastructure construction and digital empowerment to reduce emissions. Accelerate the application of digital technology in daily production and life, accelerate the realization of higher-quality interconnection, and provide solid information infrastructure support for the development of the DE, to expand the scope of benefits of the DE in a wider area, and explore the environmental regulation model of public participation to expand the role of the DE in reducing CE. Urbanization can effectively promote the green and low-carbon development of cities, so it should actively promote the development of smart cities, accelerate the application of digital technology in urbanization, and use technology to help cities reduce CE. Strengthen the popularization and application of digital finance, reduce the carbon dioxide generated by enterprises and individuals in financial activities by building financial trading platforms, promote industrial structure reform, and improve the utilization efficiency of production resources, to improve the emission reduction effect of technology enabling.Take the regional development differences as the starting point, and implement differentiated countermeasures. Aiming at the differentiated characteristics of different regions, promote technological exchanges and innovation, and enhance the production capacity of low-carbon industries. Actively play the leading role of the industry in advanced regions, conduct in-depth mining and analysis of real-time data and historical data of energy and resource sectors, improve their economy, safety, and reliability, and empower low-carbon data to promote China’s industrial transformation and upgrading, to achieve low-carbon and intelligent industrial economic development.

## Figures and Tables

**Figure 1 ijerph-19-16094-f001:**
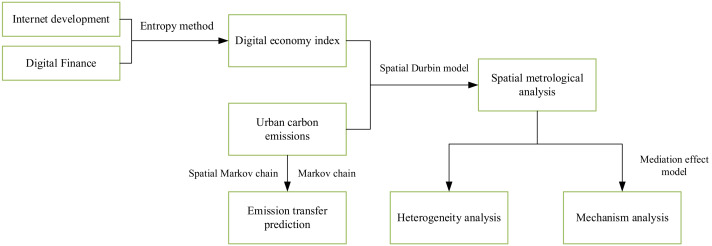
Method connection and empirical application.

**Figure 2 ijerph-19-16094-f002:**
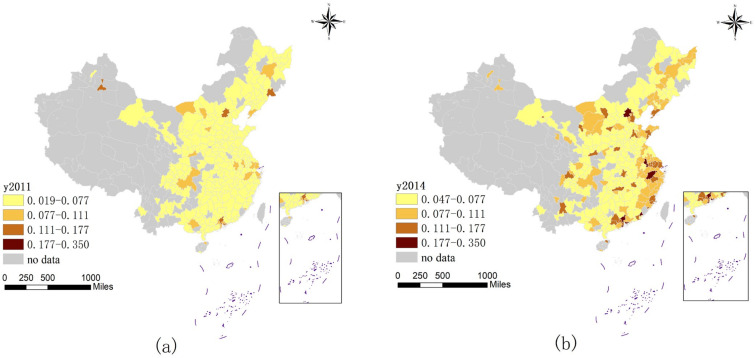
Temporal and spatial evolution pattern of DE development level in 2011 (**a**), 2014 (**b**), and 2017 (**c**).

**Figure 3 ijerph-19-16094-f003:**
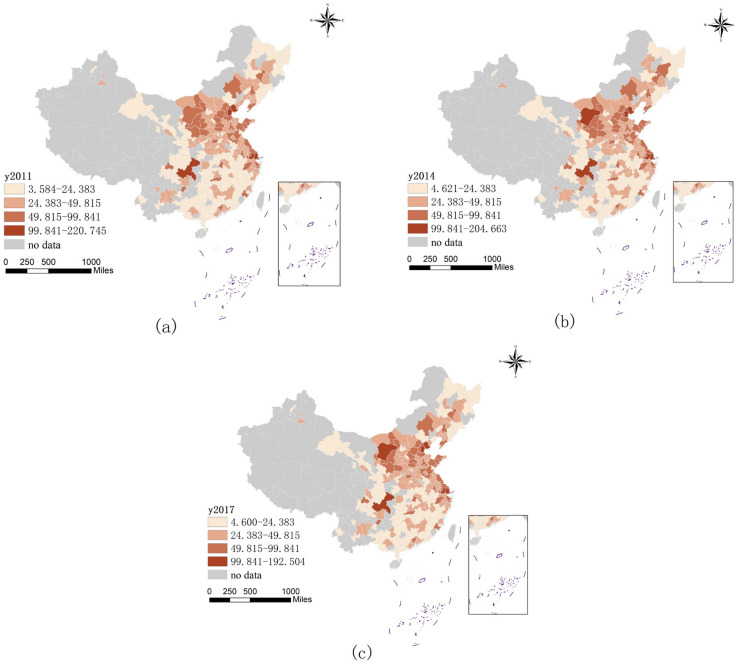
Temporal and spatial evolution pattern of CE intensity in 2011 (**a**), 2014 (**b**), and 2017 (**c**).

**Figure 4 ijerph-19-16094-f004:**
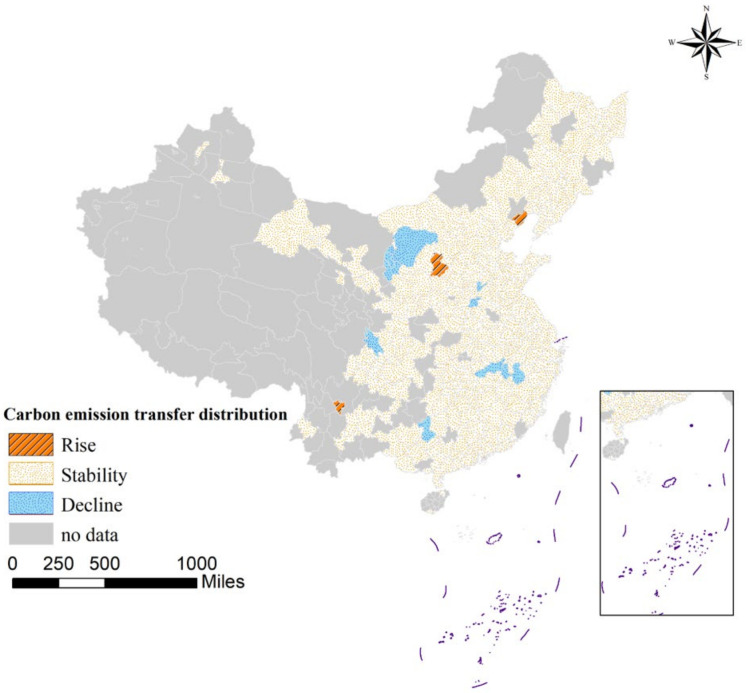
Spatial distribution of CE intensity type shift in Chinese cities.

**Table 1 ijerph-19-16094-t001:** Construction of DE indicators.

Primary Index	Secondary Index	Index Interpretation	Unit
Digital Economy development level	Internet penetration	Number of internet users	hundreds of people
	Related business output	Total telecom services per capita postal services per capita	million
	Industry practitioners	Proportion of computer services and software practitioners	%
	Mobile phone penetration	Number of mobile phone users	hundreds of people
	Digital finance development	Digital financial inclusion index	-

**Table 2 ijerph-19-16094-t002:** Variable definition.

Variable	Definition	Calculation	Unit
lnCO_2_	Urban carbon emissions	-	million tons
DEI	Digital economy development level	Calculated by the entropy method	-
lnGDP	The level of economic development	Take the natural logarithm of the ratio of the gross regional product to the total population	million
IS	Industrial structure	The ratio of output value of tertiary industry to that of secondary industry	%
FDI	Foreign investment	The ratio of foreign investment to regional gross output value	%
EC	Energy consumption	Convert all kinds of energy into standard coal and add up	tons of standard coal
UB	Urbanization level	The ratio of the urban population to the total population	%

**Table 3 ijerph-19-16094-t003:** Variable descriptive statistics.

Variable	Obs	Mean	SD	Min	Max
lnCO_2_	1855	3.180	0.722	1.276	5.424
DEI	1855	0.085	0.047	0.019	0.820
lnGDP	1855	10.680	0.555	9.219	13.06
IS	1855	1.376	0.635	0.235	8.755
FDI	1855	0.121	0.164	0.000	1.310
UB	1855	0.326	0.215	0.019	2.432
EC	1855	4.567	1.133	0.822	8.183

**Table 4 ijerph-19-16094-t004:** Moran’s I index value of CE intensity of Chinese cities.

Variables	I	Z	*p*-Value *
y2011	0.056	8.139	0.000
y2012	0.055	8.129	0.000
y2013	0.051	7.501	0.000
y2014	0.051	7.450	0.000
y2015	0.054	7.898	0.000
y2016	0.054	7.922	0.000
y2017	0.049	7.232	0.000

Note: * represents the significance level of 10%.

**Table 5 ijerph-19-16094-t005:** Markov transition probability of urban CE from 2011 to 2017.

	Sample Size	I	II	III	IV
I	798	0.977	0.023	0.000	0.000
II	526	0.023	0.970	0.008	0.000
III	230	0.000	0.017	0.974	0.009
IV	36	0.000	0.000	0.028	0.972

**Table 6 ijerph-19-16094-t006:** Transition probability of urban CE from 2011 to 2017.

		Sample Size	I	II	III	IV
I	I	440	0.977	0.023	0.000	0.000
	II	119	0.059	0.933	0.008	0.000
	III	29	0.000	0.000	1.000	0.000
	IV	6	0.000	0.000	0.000	1.000
II	I	321	0.984	0.016	0.000	0.000
	II	303	0.013	0.980	0.007	0.000
	III	129	0.000	0.031	0.953	0.016
	IV	6	0.000	0.000	0.167	0.833
III	I	35	0.914	0.086	0.000	0.000
	II	104	0.010	0.981	0.010	0.000
	III	66	0.000	0.000	1.000	0.000
	IV	24	0.000	0.000	0.000	1.000
IV	I	2	1.000	0.000	0.000	0.000
	II	0	0.000	0.000	0.000	0.000
	III	6	0.000	0.000	1.000	0.000
	IV	0	0.000	0.000	0.000	0.000

**Table 7 ijerph-19-16094-t007:** Model selection correlation test.

Test		Statistics	*p*-Value
LM Test	Spatial error	6192.473	0.000
	Spatial lag	4636.181	0.000
R-LM Test	Spatial error	1642.817	0.000
	Spatial lag	86.525	0.000
LR Test	Spatial error	178.46	0.000
	Spatial lag	282.85	0.000
Wald Test	Spatial error	125.41	0.000
	Spatial lag	141.12	0.000
Hausman		52.70	0.000

**Table 8 ijerph-19-16094-t008:** Spatial econometric regression results.

Variables	SDM		SDM		SDM	
	CO_2_	W_1_ × CO_2_	CO_2_	W_2_ × CO_2_	CO_2_	W_3_ × CO_2_
DEI	−0.078 ***	−2.092 ***	−0.129 ***	−2.018 ***	−0.243 ***	0.164 *
	(0.01)	(0.00)	(0.00)	(0.00)	(0.00)	(0.08)
lnGDP	0.078 **	0.961 ***	0.036	1.669 ***	0.552 ***	−0.806 ***
	(0.04)	(0.00)	(0.36)	(0.00)	(0.00)	(0.00)
IS	−0.161 ***	−0.161	−0.154 ***	−0.185	−0.202 ***	−0.066
	(0.00)	(0.22)	(0.00)	(0.13)	(0.00)	(0.36)
FDI	0.466 ***	−1.803 ***	0.354 ***	−4.001 ***	0.306 ***	0.667 ***
	(0.00)	(0.00)	(0.00)	(0.00)	(0.00)	(0.00)
UB	−0.515 ***	−0.136	−0.021	0.479 ***	−0.469 ***	0.183
	(0.00)	(0.72)	(0.26)	(0.00)	(0.00)	(0.42)
EC	0.416 ***	0.598 ***	0.408 ***	0.248	0.426 ***	0.018
	(0.00)	(0.00)	(0.00)	(0.15)	(0.00)	(0.71)
rho	0.520 ***		0.519 ***		0.080 *	
	(0.00)		(0.00)		(0.06)	
City FE	YES		YES		YES	
Year FE	YES		YES		YES	
Observations	1855		1855		1855	
R-squared	0.304		0.382		0.453	

Note: ***, **, and * represent the significance levels of 1%, 5%, and 10%, respectively; the values in parentheses represent *p* values.

**Table 9 ijerph-19-16094-t009:** Influence effect.

Variables	(1)	(2)	(3)
	Direct	Indirect	Total
DEI	−0.094 ***	−4.509 ***	−4.602 ***
	(0.00)	(0.00)	(0.00)
lnGDP	0.083 **	2.112 ***	2.196 ***
	(0.02)	(0.00)	(0.00)
IS	−0.161 ***	−0.500 ***	−0.661 **
	(0.00)	(0.12)	(0.05)
FDI	0.454 ***	−3.342 **	−2.888 *
	(0.00)	(0.05)	(0.09)
UB	−0.517 ***	−0.882	−1.400
	(0.00)	(0.34)	(0.13)
EC	0.423 ***	1.754 ***	2.177 ***
	(0.00)	(0.00)	(0.00)

Note: ***, **, and * represent the significance levels of 1%, 5%, and 10%, respectively; the values in parentheses represent *p* values.

**Table 10 ijerph-19-16094-t010:** Regional heterogeneity analysis.

Variables	Eastern		Central		Western	
	CO_2_	W × CO_2_	CO_2_	W×CO_2_	CO_2_	W × CO_2_
DEI	0.020	0.370	0.351 ***	−11.094 ***	−0.408 ***	−1.375
	(0.52)	(0.28)	(0.00)	(0.00)	(0.00)	(0.53)
lnGDP	−0.337 ***	−1.077	0.052	0.475	1.288 ***	0.862
	(0.00)	(0.23)	(0.35)	(0.77)	(0.00)	(0.59)
IS	0.209 ***	1.521	−0.148 ***	0.933	−0.416 ***	−1.183 ***
	(0.00)	(0.13)	(0.00)	(0.36)	(0.00)	(0.00)
FDI	0.781 ***	11.829 ***	−0.310 *	56.080 ***	1.433 *	−119.004 ***
	(0.00)	(0.00)	(0.09)	(0.00)	(0.08)	(0.00)
UB	−0.067	6.131 ***	−0.957 ***	10.007 ***	−1.254 ***	−4.503
	(0.37)	(0.00)	(0.00)	(0.00)	(0.00)	(0.11)
EC	0.486 ***	−2.515 ***	0.305 ***	0.759	0.173 ***	−0.940
	(0.00)	(0.01)	(0.00)	(0.28)	(0.00)	(0.37)
rho	−0.560		−0.850 **		−1.185 ***	
	(0.12)		(0.02)		(0.00)	
City FE	YES		YES		YES	
Year FE	YES		YES		YES	
Observations	686		819		350	
R-squared	0.257		0.382		0.458	

Note: ***, **, and * represent the significance levels of 1%, 5%, and 10%, respectively; the values in parentheses represent *p* values.

**Table 11 ijerph-19-16094-t011:** Heterogeneity analysis of urban agglomeration.

Variables	Urban Agglomeration		Non-Urban Agglomeration	
	CO_2_	W × CO_2_	CO_2_	W × CO_2_
DEI	−0.125 ***	2.056 **	0.128 **	23.288 ***
	(0.00)	(0.03)	(0.02)	(0.00)
lnGDP	−0.127 **	−2.320	0.212 ***	−5.598 ***
	(0.04)	(0.23)	(0.00)	(0.00)
IS	−0.288 ***	3.951 ***	−0.108 ***	−1.544 *
	(0.00)	(0.00)	(0.00)	(0.06)
FDI	0.608 ***	21.276 ***	0.192	12.677 ***
	(0.00)	(0.00)	(0.15)	(0.00)
UB	−0.205 **	−1.332	−1.012 ***	−7.568 ***
	(0.03)	(0.66)	(0.00)	(0.00)
EC	0.568 ***	−1.366 **	0.278 ***	−1.362 **
	(0.00)	(0.03)	(0.00)	(0.01)
rho	−0.575		−0.948 **	
	(0.12)		(0.01)	
City FE	YES		YES	
Year FE	YES		YES	
Observations	735		1120	
R-squared	0.334		0.471	

Note: ***, **, and * represent the significance levels of 1%, 5%, and 10%, respectively; the values in parentheses represent *p* values.

**Table 12 ijerph-19-16094-t012:** Impact mechanism test.

	(1)	(2)	(3)	(4)	(5)	(6)
Variables	IS	lnCO_2_	IF	lnCO_2_	UB	lnCO_2_
DEI	−0.5621 ***	−0.2569 ***	63.9710 ***	−0.1331 ***	0.0724 ***	−0.2569 ***
	(0.097)	(0.047)	(13.730)	(0.030)	(0.028)	(0.047)
IS		−0.1298 ***				
		(0.029)				
IF				−0.0008 ***		
				(0.000)		
UB						−0.3213 ***
						(0.062)
Constant	−4.8639 ***	−0.7339	90.0888	−0.0308	−0.7182 ***	−0.7339
	(0.777)	(0.491)	(74.613)	(0.445)	(0.191)	(0.491)
Observations	1855	1855	1855	1855	1855	1855
R-squared	0.240	0.443	0.254	0.438	0.290	0.443
Control	YES	YES	YES	YES	YES	YES
Bootstrap	−5.47 ***		−4.13 ***		−5.28 ***	

Note: *** represents the significance level of 1%; The values in parentheses represent standard error values. The value in bootstrap is the Z value.

## Data Availability

The data sets used in this research are available from the corresponding author on reasonable request.

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
