# Peer review of "Digital Economy and Environmental Quality: Insights from the Spatial Durbin Model"

_ijerph, 2022, doi:10.3390/ijerph192316094_

Round 1

Reviewer 1 Report

The aim of the authors to construct a digitalization index using the entropy value method is ambitious and the authors delivered the promise stated in the abstract. The results of the study are clear and the methods arriving at them are also clear.

The authors explained in a very understandable way method connection, empirical application and variables. Variables are defined and measured appropriately, but I suggest authors to include references even in the Methods part of the study. For example, at the lines 289-290 the following statement - At present, there have been plenty of studies on the influencing factors of CE intensity in the literature – should be demonstrated by naming few of the studies. The same observation is for reference to environmental Kuznets theory (lines 292-296). Adding at least one relevant reference is a must.

They also proved to master statistical analysis. However, I have some concerns about analysed data. For example, on line 335 is stated that “the CE data CEADs database” without being discussed what CEAD means. As IJERPH is an open journal, the general audience will need to know at the first glance that the database can be accessed (and refereed accordingly) at ceads.net.

There are only one main limitations of the paper: the paper does not provide limitations. Even the authors stated in the abstract that they are scant empirical studies investigating the link between digitalization and CE, I suggest the improvement of the Introduction section at least by extending the discussion about the link between digitalization and sustainable economic development.

Another required improvement for the article should come from providing more details in order to replicate the study in other regions.

The results are discussed from multiple angles at the Policy suggestions section of the article, but the authors state if they are relevant only for China or they can be applied into a wider context.

Author Response

Dear reviewer,

Thank you for your recognition of our work and careful review of our article. We have made appropriate modifications to the article. I hope you work smoothly and have good health in 2023!!!

Point 1: The authors explained in a very understandable way method connection, empirical application and variables. Variables are defined and measured appropriately, but I suggest authors to include references even in the Methods part of the study. For example, at the lines 289-290 the following statement - At present, there have been plenty of studies on the influencing factors of CE intensity in the literature – should be demonstrated by naming few of the studies. The same observation is for reference to environmental Kuznets theory (lines 292-296). Adding at least one relevant reference is a must.

Response 1: Thank you very much for your affirmation of our research. We agree with your suggestions and add relevant research literature as support(Please see line number 298 and 304).

Point 2: They also proved to master statistical analysis. However, I have some concerns about analysed data. For example, on line 335 is stated that “the CE data CEADs database” without being discussed what CEAD means. As IJERPH is an open journal, the general audience will need to know at the first glance that the database can be accessed (and refereed accordingly) at ceads.net

Response 2: Thank you for your valuable comments. We added footnotes to the source of carbon emission data for explanation and attached a link to the database(Please see line number 343 - 344).

Point 3: There are only one main limitations of the paper: the paper does not provide limitations. Even the authors stated in the abstract that they are scant empirical studies investigating the link between digitalization and CE, I suggest the improvement of the Introduction section at least by extending the discussion about the link between digitalization and sustainable economic development.

Response 3: Thank you for your valuable comments. In the introduction, we added relevant discussions on the impact of digitalization on sustainable economic development and cited relevant literature as support. Based on the relationship between the two, we introduced the theme of this study(Please see line number 55-63).

Point 4: The results are discussed from multiple angles at the Policy suggestions section of the article, but the authors state if they are relevant only for China or they can be applied into a wider context.

Response 4: In the policy recommendations section, we added the applicable objects of this study. Although China is the research object of this study, the digital economy development of many developing countries is similar to that of China, so the relevant conclusions and recommendations of this study can be applied to other developing countries(Please see line number 706-713).

Thank you so much for your careful check. The manuscript has been thoroughly revised and edited by a native speaker, so we hope it can meet the journal's standard. Thanks so much for your useful comments.

Reviewer 2 Report

A review on the manuscript in International Journal of Environmental Research and Public Health entitled „Digital economy and environmental quality: Insights from the spatial Durbin model“.

This study constructs a digitalization index using the entropy value method and spatial Markov chain, and uses the spatial Durbin model to analyze its influence mechanism and impact on urban carbon emissions in China's 265 prefecture-level cities and municipalities from 2011 to 2017.

Broad comments

Research methods have been described at satisfactory level. The conclusions are based on analysis and are adequate.

Academic writing should be objective. If it is subjective or emotional, it will lose persuasiveness and may be regarded as relying on emotion rather than building a reasonable argument based on evidence. The language or informal writing should therefore be impersonal, and should not include personal pronouns. For most subject areas the writing is expected to be objective. For this the first person (I, we, me, my, etc.) should be avoided. In this article on line 71 is written „we used“, on line 78 is written „we construct“, on line 79 is written „we explore“, on line 186 is written we first evaluated“, on line 283 is written „we selected“, on line 296 is written „We take“, on line 331 is written“we built“, on line 398 is written „we should“, on line 404 is written „we need“, on line 413 is written „we can“, on line 460 is written „we chose“, on line 616 is written“we observe“, on line 654 is written „we found“, on line 658 is written „we need“, on line 718 is written „we should“. Eliminating personal pronouns from writing is highly recommend.

The article needs a thorough technical correction.

Specific comments

References [xx] must always be preceded by a space. The references on lines 25, 27, 41, 57, 59, 62, 102, 104, 106, 109, 114, 118, 125, 134, 138, 143, 146, 152, 153, 155, 166, 169, 174, 177, 275, 283, 297, 310, 316, 544 and 612.

Line 31 is missing a space before "(CE)".

Figures consisting of several parts must be marked separately, for example (a), (b), (c) and the corresponding explanations must be provided in the caption of the figure, not in the graphic area of the figure. Figure 2 and Figure 3.

All equations must be created with the equation editor, this applies to equations on separate lines as well as in text. All equations should be aligned centrally. Equation numbers must be right-aligned and vertically centered. Lines 202, 204, 206, 209, 211, 213, 231, 249, 260, 605, 606.

Formulas of chemical elements must be correctly formatted - in line 74 "CO2", "CO2" must be written.

References to tables in the text must start with a capital letter - in line 466 "table 8" is written.

The graphic quality of the figures is quite poor.

Author Response

Dear reviewer,

Thank you for your recognition of our work and careful review of our article. We have made appropriate modifications to the article. I hope you work smoothly and have good health in 2023!!!

Response: Thank you very much for your valuable comments. We listened to your comments and redone the pictures in the article and improved the clarity of the pictures. We also adjusted the format of all formulas and the citation format of references. In addition, we have also modified other formats and first person questions one by one, hoping to meet your requirements.

Round 2

Reviewer 1 Report

The authors addressed suggestions included in the first review report.

Author Response

Dear reviewer, 
Thank you for your recognition of our work and careful review of our article. We have made appropriate modifications to the article. I hope you work smoothly and have good health!!!

Reviewer 2 Report

A review on the manuscript in International Journal of Environmental Research and Public Health entitled „Digital economy and environmental quality: Insights from the spatial Durbin model“.

This study constructs a digitalization index using the entropy value method and spatial Markov chain, and uses the spatial Durbin model to analyze its influence mechanism and impact on urban carbon emissions in China's 265 prefecture-level cities and municipalities from 2011 to 2017.

Broad comments

Research methods have been described at satisfactory level. The conclusions are based on analysis and are adequate.

Unfortunately, no recommended corrections have been made to the article.

Academic writing should be objective. If it is subjective or emotional, it will lose persuasiveness and may be regarded as relying on emotion rather than building a reasonable argument based on evidence. The language or informal writing should therefore be impersonal, and should not include personal pronouns. For most subject areas the writing is expected to be objective. For this the first person (I, we, me, my, etc.) should be avoided. In this article on line 71 is written „we used“, etc. Eliminating personal pronouns from writing is highly recommend.

The article needs a thorough technical correction.

Specific comments

References [xx] must always be preceded by a space. The references on lines 25, 27 etc.

Line 9 is missing a space before "(CE)", etc.

Figures consisting of several parts must be marked separately, for example (a), (b), (c) and the corresponding explanations must be provided in the caption of the figure, not in the graphic area of the figure. Figure 2 and Figure 3.

All equations must be created with the equation editor, this applies to equations on separate lines as well as in text. All equations should be aligned centrally. Equation numbers must be right-aligned and vertically centered.

Formulas of chemical elements must be correctly formatted - in line 74 "CO2", "CO2" must be written.

References to tables in the text must start with a capital letter - in line 466 "table 8" is written.

Author Response

Dear reviewer, 
Thank you for your recognition of our work and careful review of our article. We have made appropriate modifications to the article. I hope you work smoothly and have good health!!!

Response: Thank you very much for your comments on this study again, but we have modified your comments one by one according to your requirements for the first time, and this time we have also modified them again according to your requirements, hoping to meet your requirements.
